# Analysis of Bonding Mechanisms of Various Implants and Adhesives in Laminated Oak-Wood Elements

**DOI:** 10.3390/polym14245373

**Published:** 2022-12-08

**Authors:** Andrija Novosel, Tomislav Sedlar, Josip Miklečić, Hrvoje Turkulin, Luka Lučić, Goran Mihulja, Vjekoslav Živković

**Affiliations:** Faculty of Forestry and Wood Technology, University of Zagreb, 10000 Zagreb, Croatia

**Keywords:** implants, carbon fibre, glass fibre, aluminium, epoxy adhesives, PUR adhesives, PVAc adhesives, oak wood

## Abstract

This study analysed the bonding mechanisms and strength between wood and non-wood implants in producing laminated oak-wood beams. The suitability of different types of adhesives, namely for load-bearing and general purpose, was also analysed. Three different types of non-wood implants—carbon fibres, glass fibres, and aluminium were glued with epoxy resin (ER), thermoplastic 1-k polyurethane adhesives (PUR), structural polyurethane adhesives (PUR 2 and PUR 3), and polyvinyl acetate (PVAc) adhesives and bonds were tested for shear strength (SS) according to ISO 6238:2018. Results of the bond quality expressed as the ultimate load to failure and displacement were recorded using the universal mechanical testing machine in combination with the digital image correlation (DIC) method. Before the shear test, all the samples were conditioned in dry and wet climatic conditions. Test results indicated that the application of PUR adhesives for bonding carbon and glass fibres with oak wood could sufficiently replace two-component ER, which is generally recommended for such purposes but is very challenging to utilise in industrial conditions. PVAc adhesives proved efficient only for combination with AL implants and in dry conditions. Aluminium sheets were shown to require surface pre-treatment, such as sanding and degreasing or a different type of adhesive to achieve sufficient adhesion.

## 1. Introduction

Wood is widely applied as a structural building material, but it also finds its use for non-structural building components (windows, doors, curtain walling, etc.). However, considering changing construction trends and requirements in contemporary architecture (large spans, open spaces, slim window scantlings with cross-sections as small as possible but with supreme mechanical properties), wood as a building material has limitations. To bridge these limitations or improve its favourable properties, adhesive bonding in lamination and additional reinforcements of wood are essential in obtaining high quality wood products, especially for optimising production costs and consumption of raw materials in the wood construction industry.

Fir and spruce are commonly used wood species for structural elements. They are often laminated or/and reinforced to bridge large spans. Softwoods that are amply present in Europe have advantages over hardwoods in as much as they are easy to process, are affordable and low in weight. In addition, they may have up to 38% better thermal insulation properties [1]. Many studies focused on improving the mechanical properties of glued laminated timber by combining wood material of specific properties with additional non-wood materials, thus reducing the overall cross-section. Various materials and designs of products for reinforcements have been studied, such as natural fibre materials [2,3], steel bars and profiles [4,5,6], glass fibres (glass fibre reinforced plastics, GFRP) and carbon fibres (CFRP, i.e., carbon FRP), or plates [7,8,9,10,11,12]. Generally, the wood species investigated in such attempts were softwoods, the beams were of large dimensions, and the reinforcements were often visible. However, there were also attempts to study the reinforcement effect on structural elements of smaller cross-sections [13] and the introduction of non-visible structural implants [14,15].

Using hardwoods instead of softwoods opens the option of applying the species with better biological durability and desired aesthetic appearance, yet with leaner elements, smaller cross sections, and lower material consumption. According to former investigations on the bonding performance of hardwoods [16,17], the gluing efficiency and mechanical properties of load-bearing elements from beech, ash, and oak have a promising potential for use in glulam and other timber products with enhanced performance.

However, the compatibility of various non-wood materials and commercial adhesives is not yet fully understood, and the strength and durability of such bonds are not adequately addressed. Polyvinyl acetate (PVAc) adhesives, widely applied for non-structural uses due to their affordable price and good bond properties, have been scarcely tested in combinations with non-wood materials used as reinforcements [1] and their glue-bond strength and durability is therefore still not assured. On the other hand, the general recommendation for gluing CFRP, GFRP or aluminium with wood is using 2K epoxy resins which are relatively expensive, health hazardous, and technologically impractical in industrial processes. Polyurethane (PU) adhesives, particularly single-component variants, have been used for glulam for nearly two decades and are standardised (EN 15425 [18] and related standards). These formaldehyde-free and solvent-free adhesives are advantageous as they give invisible joints, render strong joints with good bond quality and durability, and are fast-curing at room temperature [19,20,21]. However, their ability to be used with aluminium, GFRP or CFRP is still not fully proven and is, therefore, the focus of this research. 

According to [17], surface preparation is a significant issue affecting the gluing quality of engineered wood products. The surface structure prepared/processed for gluing is essential in good wetting, which is vital for determining the adhesion ability [22]. Wetting may be problematic with hardwoods due to their high density, very smooth surface and presence of extractives. However, good wetting allows for better adhesion forces on a molecular level. In contrast, mechanical interlocking on the structural level between the glue layer and the wood surface may result in improved bond performance [23]. Furthermore, an increase in surface roughness, to some extent, enhances wettability, and as a result, better liquid spreading by capillarity can be obtained. 

Planing is the dominant surface machining technique in the production of engineered wood, but the alternative method for surface processing could be face milling [24]. In face milling, the tool cuts perpendicular to the fibres, thus opening the structure for better penetration of the adhesive, still leaving an even, smooth surface. Such increased micro-roughness of wood is supposed to allow for a larger specific surface, better molecular interactions (evidenced in higher free energy, i.e., the lower contact angle) and in the anchoring effect of the set glue in the substrate. Generally, the surfaces processed by planning exhibit somewhat poorer wetting properties than those processed by face milling. Therefore, we opted for face milling as a technical parameter which should result in good bonds between the implants and oak wood. 

The primary motivation of this study was to investigate the bonding mechanism and joint strength properties between hardwood, which is more demanding for long-lasting glue bonds than softwoods, and notoriously inert materials such as metal for wood glued joints. Hence, the combinations of different non-wood materials to be used as reinforcements, wood adhesives, and oak wood as a substrate were tested with a focus on the glue bond quality.

## 2. Materials and Methods

Three types of materials were combined to form bonds to be tested: oak as the substrate—the primary material, non-wood materials for implants (carbon fibre, glass fibre, aluminium), and adhesives. The main elements of the investigation were:The bonding mechanism and quality of oak wood with three different non-wood materialsThe applicability of different types of adhesives and their bonding performance.

### 2.1. Wood

Oak wood (*Quercus robur* L.) was selected for this study. The average density of tested oak wood at 9% moisture content was 687 kg/m3, equivalent to relevant data for natural oak wood [25]. All selected oak-wood elements were straight grained and free from visual defects or knots. Kiln-dried elements were stored in a conditioning room at 23 °C and 50% relative humidity until they reached 9 + 0.5% of equilibrium moisture content (EMC). 

After conditioning, the boards were sawn into 60 × 12 × 300 mm elements. Next, each element was machined (processed) by face milling to a total dimension of 60 × 10 × 300 mm. Before gluing, the wood surface was cleaned from dust and dirt. Next, elements underwent a surface roughness measurement and a wettability test.

The sample preparation process (material with adhesive only or with various implant layers, upper left), cutting scheme with the removal of the edge (boundary) portions, and the geometry of samples (mm) are presented in Figure 1. 

The face milling was carried out with a cutting depth of 2 mm, a feed speed of 5 m/min, and a rotation frequency was 20,000 r/min. The total diameter of the tools was 80 mm with a blade rake angle of 37°.

Surface wettability tests were performed 2 h after mechanical surface processing. Wetting of the surface occurs when the contact angle is smaller than 90°—the lower the contact angle, the better the wettability. The surface energy of oak-wood was measured using contact angles of water, diiodomethane and formamide (according to the Owens-Wendt-Rabel-Kaelble (OWRK) model [26]) by sessile drop method and goniometer OCA 20 (temperature 23 °C, ten seconds after applying the liquid drop). In our case, the contact angle (avg and (std)) for a group of 15 readings amounted for water (48.67° (5.90)), diiodomethane (25.00° (2.83)) and formamide (15.43° (3.93)), and surface energy total (60.53° (4.41)) m/Nm, dispersive component (44.05° (2.14)) m/Nm and polar component (16.48° (6.48)) m/Nm.

These values of contact angles and free energy indicate the usual range for freshly processed oak-wood, indicating no surface deposit of extractives or excessive surface exposure to ageing or oxygenation. However, the wettability for face-milled surfaces proved somewhat better than the values reported for planed oak surfaces [27].

Roughness measurements of surface processing were performed with a commercial mechanical profiler Mitutoyo SJ 500. (stylus tip, angle 90°, radius −10 um, speed of 1 mm/s). The surface roughness was measured on seven samples prepared by face milling along the fibres (radial wood surface structures). Description of roughness parameters used in this research (average roughness—Ra; the max height of profile—Rz; kurtosis coefficient—Rku and skewness coefficient—Rsk) was calculated/interpreted according to ISO 4287:1997 [28]. 

The samples had the following characteristics: Ra: 10.65 μm; Rz: 79.77 μm; Rku: 6.222 μm, and Rsk: −1.26 μm (average value), equivalent to relevant data for face milling [24].

### 2.2. Selection of Adhesives

The adhesives used in the study were selected according to the producer’s recommendation for gluing non-wood materials to wood (ER) and for the standard adhesives used in the wood industry for load-bearing products (1K PU and 2K PU) and non-bearing purposes (PVAc). Details about the adhesives and gluing technology are presented in Table 1. 

### 2.3. Non-Wood Materials

Carbon fibres, glass fibres, and aluminium plates were used as implants between wood elements. The types of implants used in this study were a cross-woven (plain) carbon fibre sheet with a density of 200 g/m^2^, a glass fibre sheet with a density of 166 g/m^2^, and a 5 mm thick aluminium plate. The properties of inserted implants provided by manufacturers are presented in Table 2. 

Before installation into wood samples, aluminium plates were sanded with sandpaper P120 and without any other pre-processing. Dust from sanding and possible traces of grease were cleaned with acetone. Adhesives were applied on both wood surfaces by a serrated plastic applicator (depth of notches depending on implant type) and glued over implants into sandwiches.

### 2.4. Preparation of the Test Pieces

Standard-sized samples are presented in Figure 1 according to ISO 6238:2018 [30]. The oak-wood lamellas (60 × 10 × 300 mm) were glued together under 1 N/mm2 pressure for four hours. After that, the glued assemblies were conditioned for ten days in a standard climate of 23 °C and 50% r.h. (Figure 1). Moreover, finally, the beams were sawn out at dimensions 25 × 300 mm, and the specimens were processed (Figure 2). In total, 658 samples were prepared for investigation. Five hundred seventy-two samples were exposed to the standard climatic environment and 86 to climatic cycles of dry and humid environments. Reference samples served for comparison of the mechanical properties of pure wood laminates and those with implants. 

The amount of adhesive was applied according to the producer’s recommendation and was different for samples with implants because of the different fibre thicknesses and structures. 

Gluing was conducted under laboratory conditions (23 °C/50%) within 12 h of processing the wood surfaces. Aluminium surfaces were sanded (grit size 120) and degreased with acetone. Adhesives were applied with serrated trowels according to the general manufacturer’s recommendations but with different trowel notch sizes, depending on the viscosity of the adhesive, contours of the implant surface and desired glue spread rate. The pressure was applied for adequate time to achieve the complete curing of adhesives. During the curing of adhesives, the carbon and glass implant sheets were axially stressed in tension by 20 N loads to produce pre-stressed assemblies. Further conditioning of the set assemblies in the standard climate took ten days.

### 2.5. Test Setup

Glue bonds were tested for shear strength (SS) in compression according to ISO 6238:2018 [30] in standard laboratory conditions (23 °C/50% relative humidity). 

Before testing, the standard-sized specimens were machined from the glued assemblies and subjected to climatic conditioning to test their resistance to humidity changes. Specimen marked as “st” were subjected to treatment A, and samples marked as “ptr” were initially subjected to treatment A, then to 3 cycles of dry and humid conditions (marked as B and C), and finally conditioned at A o reach equilibrium moisture content of wood. The details of treatments are further explained in Table 3.

Unreinforced oak-wood laminated elements glued with each adhesive served as a reference (code REF) for comparing bond strength, displacement, and the size and nature of deformations of specimens during loading. 

### 2.6. Testing and Measuring System

Testing was executed on the 10 kN universal testing machine with the constant rate of traverse set at ca 5–7 mm/min, depending on the type of bond according to ISO 6238:2018 [30]. The specimens reached the ultimate loads within 60 to 80 s, which agrees with the requirements of the mentioned standard. Loads were recorded using the standard Trapezium programme and were combined in software with displacements recorded optically.

The strain and deformation measurements were conducted using the digital image correlation (DIC) method using a set of two 12 M cameras to enhance the accuracy of the measurement. A spotted random pattern with white and black colours was applied to the specimen’s side surface for the application of the DIC method (Figure 3). 

The system captures eight images per second during the loading process. The software analysed the whole series of recorded images as the correlation between the displacement, and equivalent mises strain to describe deformations. To describe deformations, the technical strain was used as the quotient of the length change and reference length.

The software recorded the relative deformations of the specimens during the loading phase. This was enabled through the parallel record of the displacements (strain, i.e., deformations) at the corresponding load. In such a way, the system gave insight into the distribution of deformations and the toughness of particular wood-implant combinations, stress concentrations due to the rigidity of the implant or the glue line, and consequently, the conclusions about the efficiency of the reinforcing system.

The results are presented either in the form of the load–displacement curves or the shear strength values (ultimate load to failure per glued surface), along with the wood failure proportion or the strain distribution profiles. The latter should also give additional information about the differences in deformation responses between the standard samples and those that underwent the climatic condensation pre-treatments.

Statistica software (TIBCO Data Science Workbench version 14.0.0.15) was used to compare and analyse the results. ANOVA method (Levene’s test, post hoc Tukey) was applied for non-treated groups of samples. In contrast, the results of climatically pre-treated samples were analysed using the non-parametric Kruskal–Wallis test because of the smaller number of specimens. 

## 3. Results and Discussion

### 3.1. Load—Displacement of Reference Samples (Wood to Wood)

The average results of ultimate load to failure (ULF) and the corresponding displacement of the specimens are shown in Figure 4. The presented results are values of the reference model (wood to wood) and their representative specimen glued with different adhesives. These results were used to compare the implant and adhesive effects.

Figure 4 demonstrates the strength values for conditioned samples in excess of 9 N/mm^2^ (min value for 0.5 mm thick glue line of structural PUR adhesives in EN 15425 [18]), reaching even values as high as 15–16 N/mm^2^ for PUR 1, PUR 2 and PUR 3 adhesives. ER values are comparable to those of PVAc adhesive (see the fourth column in Figure 5a) and confirm the results by [31]. Therefore, PVAc adhesive would also satisfy the requirements of EN 204 [29] (11 N/mm^2^ > minimum required 10 N/mm^2^). On the other hand, Figure 4 indicates different toughness of particular bonds in standard conditions since the strain (deformation) values, although small, clearly demonstrate up to twice greater strains for PUR 2 and PVAc adhesives than ER and PUR 1. The differences are discernible in the pure elastic region (the line is drawn at 5.0 kN load), where the deformations of PVAc and PUR 2 bond surfaces reach nearly 45–90% greater values than rigid PUR 1 and PUR 3 bonds.

After stressing the specimens through swelling and shrinking in the climatic pre-treatment cycles, the differences between adhesives become more apparent. Some adhesives (PVAc and PUR 2) exhibit lower strength values than in the standard state but still a great bond toughness. However, the deformations of PUR 2 bonds are more significant after cycling treatment than in standard conditions. On the contrary, rigid ER and PUR 3 bonds exhibit minimal strength decrease after the pre-treatment cycling, but their deformations become even lesser than in the standard conditioned state. Neither the stress relaxation in wood during climatic cycling nor the newly introduced stresses on glue bonds through swelling and shrinking affect the strength and toughness of rigid ER and PUR 3 bonds, which may have implications for the behaviour of implant bonds with such adhesives.

### 3.2. Shear Strength of Standard-Conditioned Implant Bonds

There was no statistically significant difference between the bond shear strength of samples reinforced with the aluminium implant, glued with ER and PVAc (higher strength), and those glued with PUR 1 and PUR 3 adhesive (lower strength). The same is valid for CRFP implant bonds glued with ER and PUR 3 adhesive (see the circles in Figure 5a). PVAc adhesive yields significantly weaker bonds with CFRP and GFRP implants compared to other adhesives. If we compare the strength values of implant bonds with REF wood samples, we can notice two groups of similar values. One group forms all PUR wood–wood bonds (blue circle in Figure 5a), and they are not significantly weaker than the bonds with CFRP and GFRP implants (see also the circle in Figure 5b). Epoxy, on the contrary, exhibits even better bond strength values with CFRP and GFRP implants than in wood-to-wood bonds, whereas it is poor for aluminium bonds. PVAc adhesive is suitable for gluing aluminium to wood but insufficient for other implant bonds. Figure 5b confirms this statement, showing that aluminium is a problematic substrate for gluing implants to wood with any other type of adhesive.

### 3.3. The Effect of Climatic Stressing on Bond Strength

The results for wet reinforced samples showed that the effect of implants makes some significant difference in SS. To analyse the results, we used Kruskal–Wallis tests, multiple comparison *p* values. For ERptr adhesive test were H (3, N = 18) = 12.26316 *p* = 0.065, for PUR 1 ptr were H (3, N = 15) = 9.22333 *p* = 0.0265, PUR 2 ptr were H (3, N = 20) = 11.61714, *p* = 0.0088, for PUR 3 ptr samples were H (3, N = 15) = 0.0000 *p* = 1.000, and for PVAcptr adhesive were H (3, N = 18) = 9.963158 *p* = 0.0189. (H—Kruskal–Wallis test reference range, N—number of samples).

All adhesive strength values on aluminium implants are relatively weak and generally further weaken after the climatic pre-conditioning. PUR 3 bonds actually fell apart after cyclic pre-treatment, even before testing. On the other hand, the PVAc adhesive would appear to give good bonds on the AL surface. However, the extensive scattering of these results leads to a general conclusion about the inadequacy of the PVAc-AL bond, particularly seeing graphs in Figure 6, where it will be indicated that most of the PVAc-AL failure occurred through adhesion loss on the AL surface. PVAc adhesive exhibited low bond quality on CFRP and GFRP materials, again with adhesion mode of failure (Figure 6e) and very low strengths, which renders PVAc generally unsuitable for reliable gluing of any type of implant. Climatic pre-conditioning might have caused two antagonistic effects. On one side, the moderate swelling and shrinking of wood might have caused relaxation of stresses within the bonds and actually improved the strength values and wood failure proportion. On the other side, shear stresses due to significant swelling and shrinking would additionally weaken the bond. The graphs in Figure 6a–d indicate that the strength values of ER and PUR adhesives on CFRP and GFRP implants did not change much and remained in the same strength range. However, the scattering of results is notably lower after three pre-conditioning cycles, indicating that our climatic stressing system did not cause excessive shear forces on the bond line while reducing some internal stresses of the bonds. In that sense, more intensive water action (waterlogging or even boiling) would be needed to test the various implant bonds with selected adhesives for their durability, strength and resistance to exterior conditions.

### 3.4. Failure Modes

Wood failure (proportion of a fractured area within the wood substrate) is a very common appearance in well-bonded joints. The fracture within the wood substrate is desirable for glued elements, indicating that the bond strength (a composite of the adhesive, interlayer, and adhering wood) is better than the cohesive strength of the wood substrate. In the case of dominant wood failure, the adhesive strength between the elements remains unknown, but it can be assumed that the overall bond strength is higher than the cohesive strength of wood.

Determining the wood failure percentage is very important in defining the quality of the adhesive bonding [32], and the estimate of the fractured wood or adhesive area was conducted following the provision of the standard ISO 10365 [33]. However, to prevent the main problem of this method (subjectivity and different visual/professional abilities of examiners), the mode of failure was independently determined by two different people. After testing the shear strength of the bonds, the proportions of surfaces were selected to be the cohesive failure within wood (WF), including a thin layer of fibres remaining on the glue line), the cohesive failure within the glue line (CA) and the adhesion failure, either distinctively between two jointed surfaces (adhesive and substrate) or within the thin intermediary layer with only a few, separated fibres remaining on the adhesive layer (A).

Figure 7 demonstrates the wood failure for representative mark st—standard conditioned samples of each implant group and adhesives, and Figure 8 and Figure 9, failure proportions of st—standard conditioned samples and ptr—climatically pre-treated samples.

ER proved to be rigid and, surprisingly enough, relatively weak in cohesive strength because, in dry condition (Figure 8), almost 80% of failure of wood-to-wood (REF) samples happens in a cohesive type within the glue line (also indicating the reference to the photograph/micrograph of the specimen). After climatic conditioning (Figure 7 and Figure 9), the situation is not changed much, although the share of the substrate cohesive failure increased a bit. However, when implants are glued with ER, the failure is always dominantly cohesive in the substrate (wood failure).

PUR 1 and PUR 2 gave strong wood bonds, with almost complete wood failure in REF samples (Figure 7 and Figure 8). Strong bonds were also obtained on CFRP and GFRP samples, although with a more significant proportion of adhesive (glue-to-implant) failure. On aluminium implants, however, the bonds were weak, primarily due to the sheer adhesive bond strength of the glue line to the aluminium surface. The pre-conditioning did not change much, either the bond strength, which remained high or the nature of the failure (adhesive failure of the glue to the fibre matrix of the implant). The adhesion to wood was more excellent than to fibres, but both adhesive strengths remained high.

PUR 3, being of lower viscosity, penetrated the fibres better, which resulted in similar strength values, but in a significantly more significant proportion of wood failure than with PUR 1 and PUR 2 adhesives. However, shear stresses due to swelling and shrinking could not influence the failure mode to a large extent, so the pre-treated bonds were generally the same as those of standard-climatised specimens. 

PVAc failure modes confirmed the poor strength of all implant bonds with this adhesive: the failure modes were predominantly adhesion type with all implant materials. 

Aluminium proved to be an unsuitable material to glue with wood adhesives, giving mostly adhesion failures and poor bond strength (see also Figure 5a).

The failure analysis gives more information about the bond quality than the pure strength values. These two indicators should, however, always be observed simultaneously to understand and interpret the quality of the bond. For example, the comparison of PVAc and PUR 2 failure modes shows mostly an adhesive failure in both cases. Still, the strength values of PVAc were incomparably lower, obviously because of the poorer PVAc adhesive strength to the fibres.

### 3.5. Axial Deformation of Dry and WET samples

Figure 10 indicates the deformations of specimens, as recorded by digital image correlation. One representative specimen per sample was chosen to illustrate the behaviour of the bond. The images were taken at the load of 7 kN, i.e., at the stressing level very close to ULF of the reference (wood-to-wood) ER and PVAc samples. Red colour indicates the most significant deformations, i.e., the zones of maximum strain within the probes. At the same time, the red colour may also indicate the stress concentration areas during testing in specimens with rigid bond lines (at the edges of the glued surfaces).

It can be seen that PVAc bonds exhibited the most significant deformations with all implants except with aluminium, which was expected in accordance with PVAc’s notorious viscoelasticity. However, the aluminium implant, glued with a very thin PVAc layer and other adhesives, proved too rigid to show any strain.

The distinction between PUR adhesives may be seen in wood-to-wood bonds since PUR 3 bonds proved rigid, whereas PUR 1 and PUR 2 yielded a bit at this loading level. Implants, however, contributed to the rigidity of the joints, and all PUR adhesives, similarly to ER, showed small deformations, mainly concentrated at the edges of glue lines, where theoretical stress concentrations should be the greatest in any case.

After the climatic cycling, the deformations can be seen as generally greater, particularly at the edges of bond lines. This confirms the assumption that the bonds became less rigid due to some stress relaxation caused by moderate swelling and shrinking. However, although regular, these differences are of negligible magnitude, which was also seen in the results of strength values. However, slightly greater edge deformations of PUR 1 and PUR 2 bonds (in comparison with ER and PUR 3 adhesives) may corroborate the results of the best strength values of these adhesives (Figure 6) and the conclusion about their good bond properties with the fibre implants.

## 4. Conclusions

The attempt to seek an alternative to epoxy resins in commercial polyurethane wood adhesives for gluing the reinforcements or implants into wood laminations proved successful. PVAc adhesive, commonly used for non-structural purposes, showed no promising results. On the other side, the single-component structural PUR adhesives, either as a standard or a fibre-reinforced product, yielded strong bonds of carbon- or glass-fibre implants (CFRP or GFRP) to wood that may be comparable or even superior to ER-bonds, with better consistency of results. Two-component PUR adhesive gave similar bond strength values but proved more rigid, which resulted in a more significant proportion of wood failure than with 1K PUR adhesives. The effect of bond rigidity on its durability in hardwood laminations should be further investigated. 

Both CFRP and GFRP materials, when introduced into oak-wood laminations, enabled equal or even better bond strength values as with wood-to-wood bonds. This finding encourages using fibrous materials to strengthen laminated structural components, even when made of dense hardwood species. Swelling and shrinking of oak wood, significantly higher than that of softwoods, did not considerably influence the bond strength as long as the bonds were exposed to moderate climatic cycles (30–70% r.h. at ambient temperatures). It seems that the positive effect of climatic cycling on the stress relaxation of bonds was more pronounced than the negative influence of sheer stresses induced by dimensional movement.

Aluminium proved challenging to glue with PUR adhesives, resulting in lower bond strength than fibrous implants and prevalent adhesion failure. Obviously, aluminium surfaces must be modified differently to enable such implants in structural wood laminations.

## Figures and Tables

**Figure 1 polymers-14-05373-f001:**
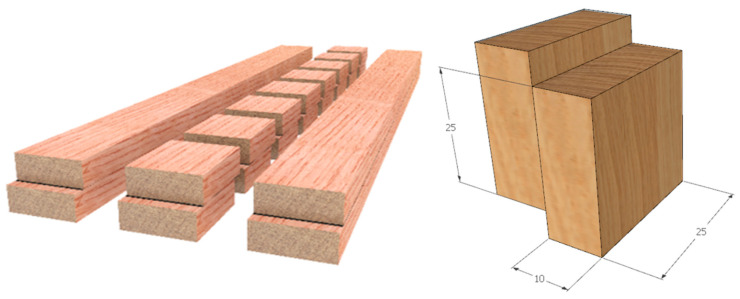
The preparation of material assemblies and the geometry of test specimens according to ISO 6238:2018. All dimensions are in mm.

**Figure 2 polymers-14-05373-f002:**
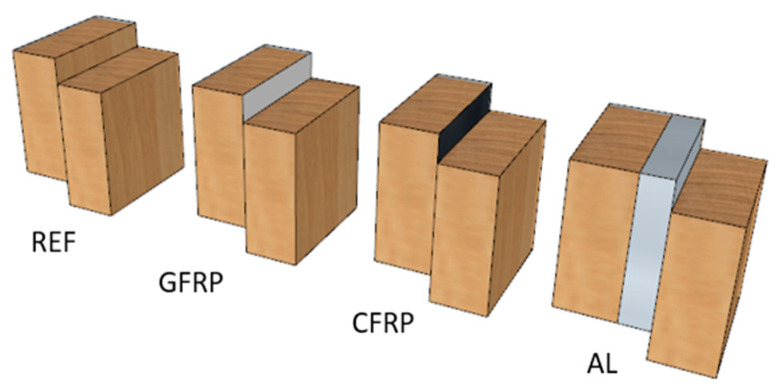
The Configurations of specimens (reference samples wood–wood, GFRP: glass fibre implant, CFRP: carbon fibre implant, AL: aluminium implant).

**Figure 3 polymers-14-05373-f003:**
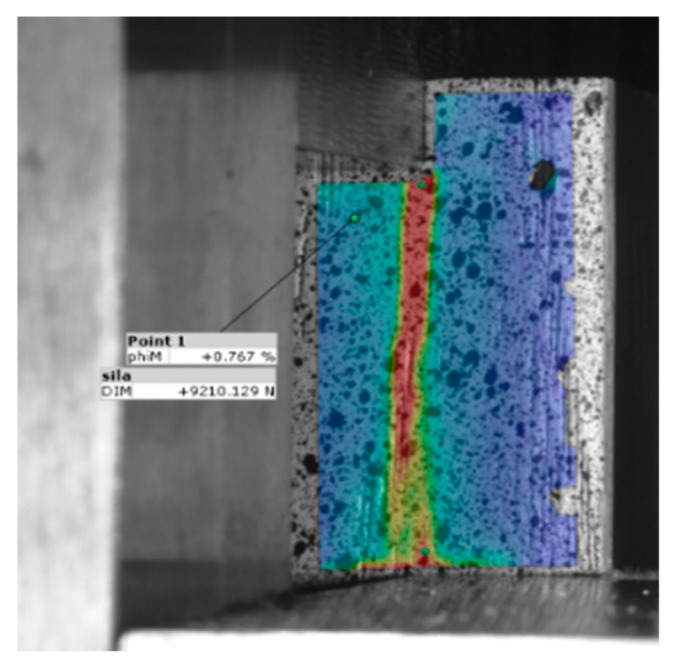
Side view of the specimen during a test with the randomly sprayed black specks pattern for digital image analysis and overlapping coloured strain distribution obtained by the software.

**Figure 4 polymers-14-05373-f004:**
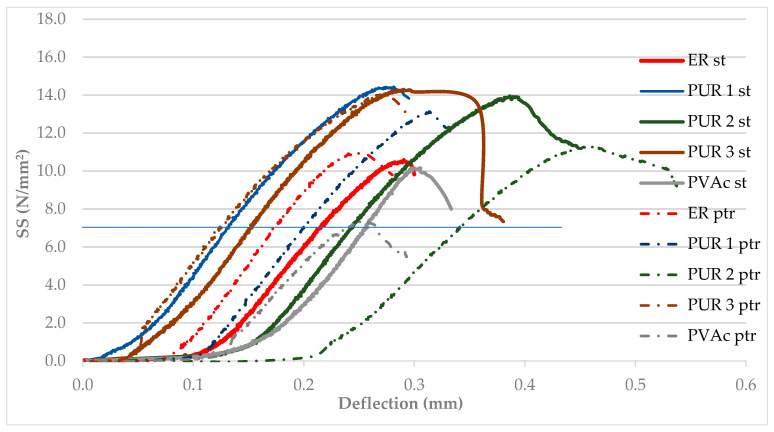
Strain—deflection curve for the reference and treated samples (mark st—standard conditioned samples, mark ptr—climatically pre-treated samples).

**Figure 5 polymers-14-05373-f005:**
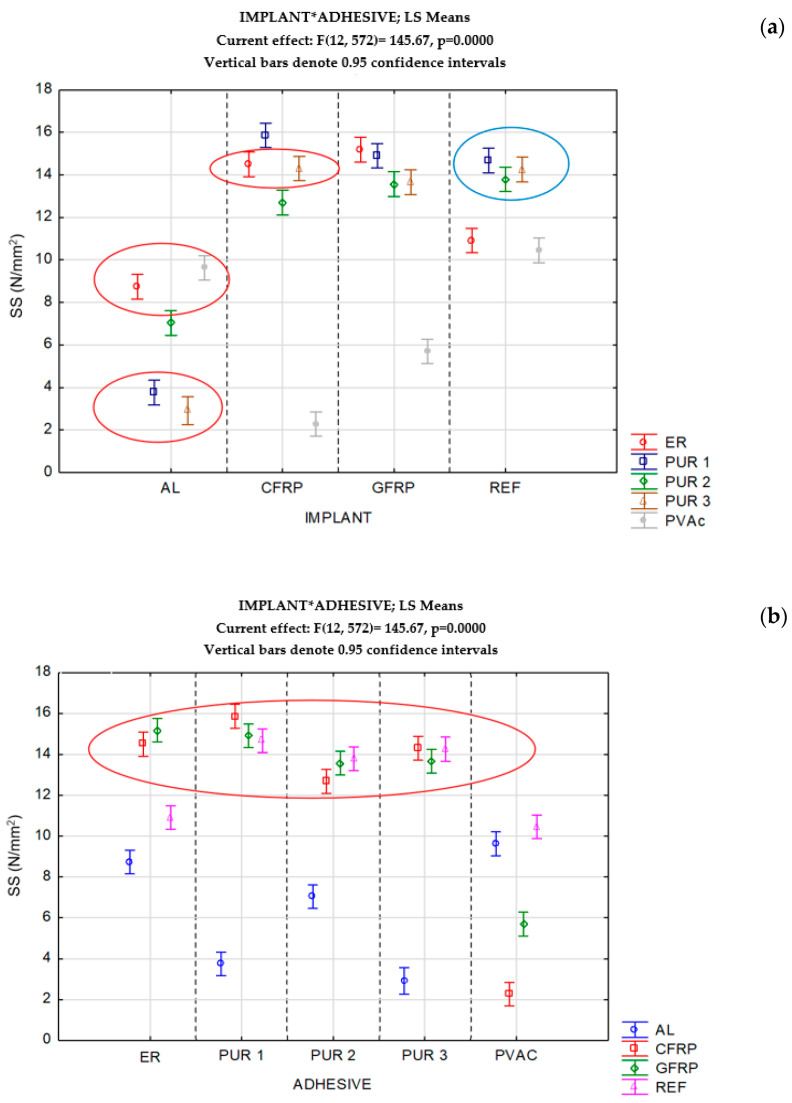
Bond effectiveness of different combinations of implants (**a**) and adhesives (**b**) in standard conditions.

**Figure 6 polymers-14-05373-f006:**
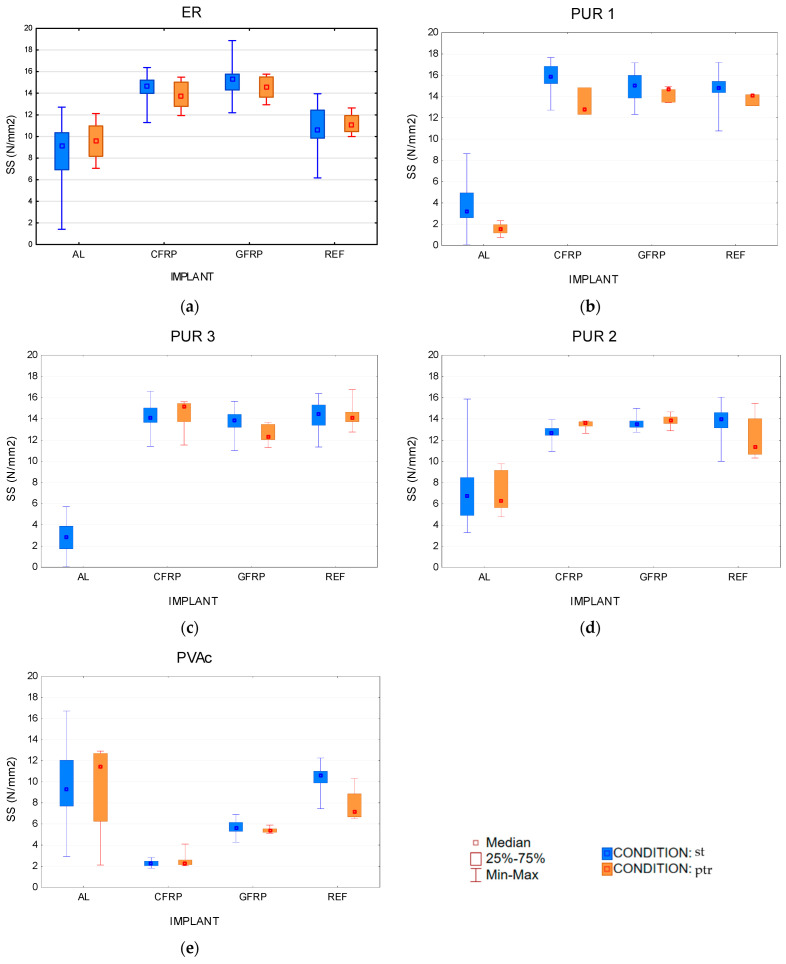
Box plot by implant group of shear strength (SS) for st—standard conditioned samples and ptr—climatically pre-treated samples. (**a**) 2-k epoxy resin, (**b**) thermoplastic 1-k polyurethane adhesive, (**c**) structural 1-k polyurethane adhesive, (**d**) structural 2-k polyurethane adhesive, (**e**) 1-k polyvinyl acetate.

**Figure 7 polymers-14-05373-f007:**
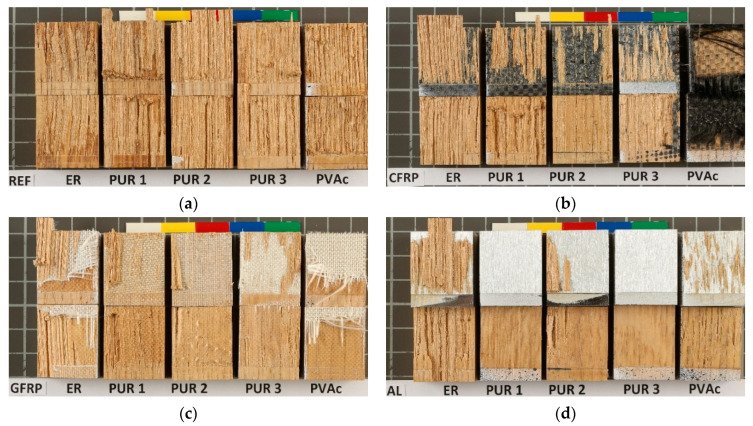
Failure mode of standard climatised samples (**a**) reference; (**b**) CFRP; (**c**) GFRP; (**d**) aluminium). (One square in the background surface is equivalent to 100 mm^2^).

**Figure 8 polymers-14-05373-f008:**
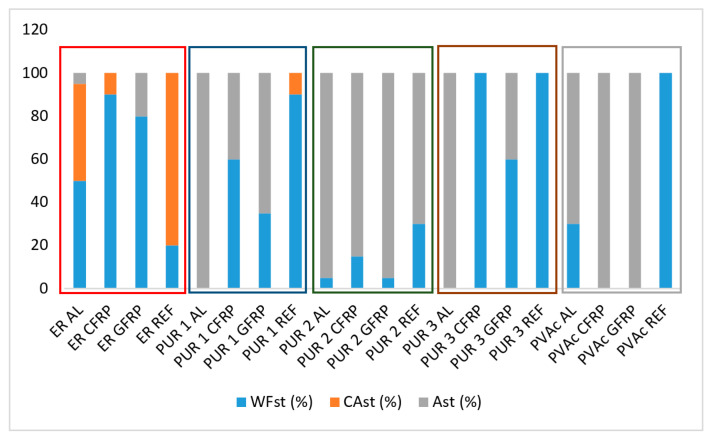
Failure proportions of standard climatised samples (WF—wood failure (cohesive), CA—cohesive failure in glue line, A—adhesion failure).

**Figure 9 polymers-14-05373-f009:**
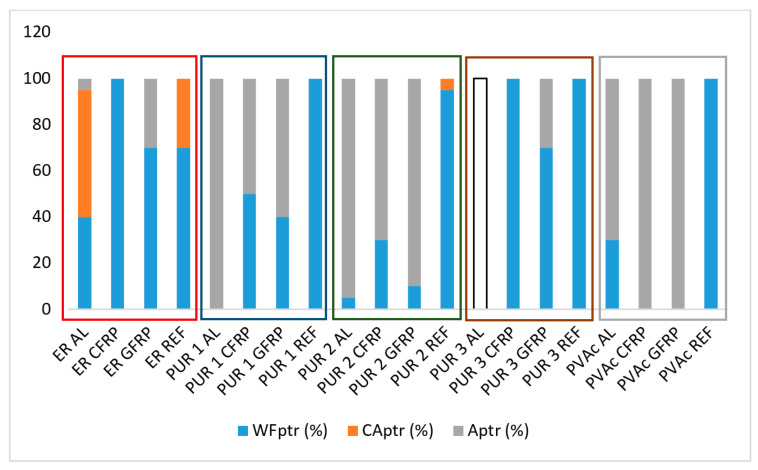
Failure proportions of climatically pre-treated samples (WF—wood failure, CA—cohesive failure of adhesive, A—adhesion failure).

**Figure 10 polymers-14-05373-f010:**
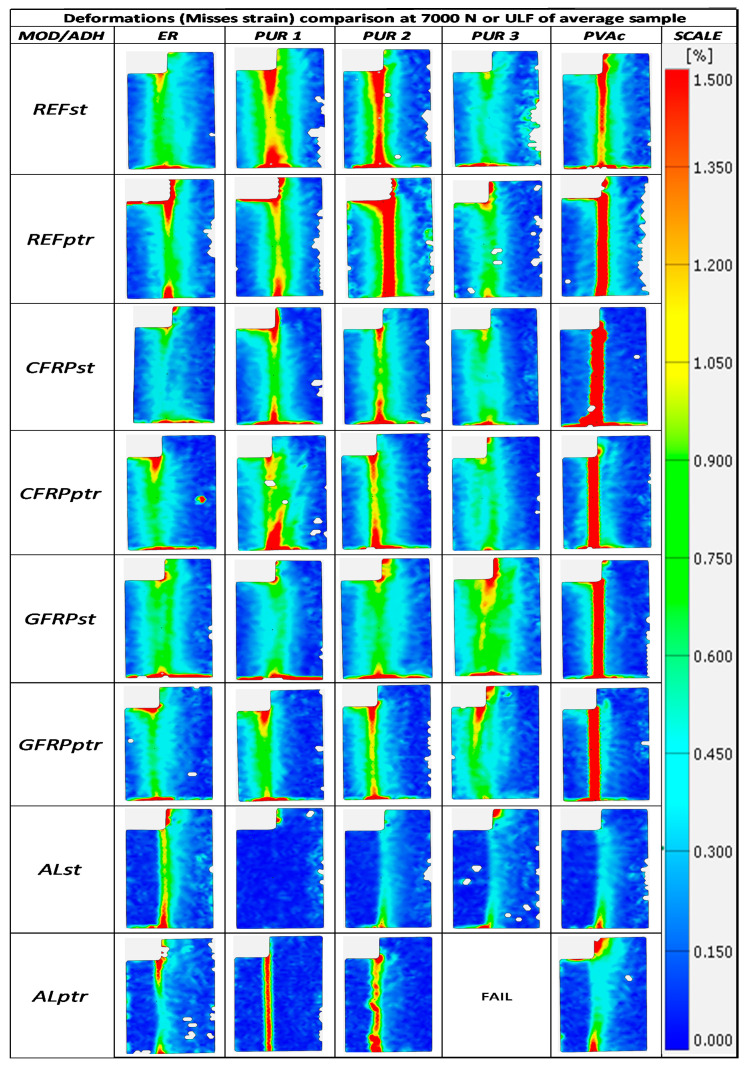
Relative axial deformations of representative samples at 7 kN which corresponds to near failure load of reference wood-wood ER samples recorded by digital image correlation. (Samples with red dot are at ULF).

**Table 1 polymers-14-05373-t001:** Technical properties of applied adhesives.

Adhesive	Technical Properties			Samples	Gluing Technology			
	Composition/Class	Density (g/cm^2^)	Viscosity (mPas)		Total Application Rate (g/m^2^)	Spread on Wood	Pressure (N/mm^2^)	Duration (h)
**PVAc**	1 k, polyvinyl acetate D4 (EN 204) [29]	1.13	7000	REF	193.3	one—side	1	4
				CFRP	233.3	both sides	1	4
				GFRP	213.3		1	4
				AL	193.3		1	4
**ER**	2 k epoxy resin + hardener 3.3:1 *w*/*w*	1.13	1800	REF	193.3	one—side	1	4
				CFRP	233.3	both sides	1	4
				GFRP	213.3		1	4
				AL	193.3		1	4
**PUR 1**	1 k Polyurethane Type II A1 (0.5 mm bond line) (EN 15425) [18]	1.13	8000	REF	193.3	one—side	1	4
				CFRP	233.3	both sides	1	4
				GFRP	213.3		1	4
				AL	193.3		1	4
**PUR 2**	1 k Polyurethane Type II A1 (0.5 mm bond line) (EN 15425) [18]	1.2	7000	REF	193.3	one—side	1	4
				CFRP	233.3	both sides	1	4
				GFRP	213.3		1	4
				AL	193.3		1	4
**PUR 3**	2 k Polyurethane Type II A1 (0.5 mm bond line) (EN 15425) [18] 2.2:1 *w*/*w*	1.1	9000	REF	193.3	one—side	1	4
				CFRP	233.3	both sides	1	4
				GFRP	213.3		1	4
				AL	193.3		1	4

**Table 2 polymers-14-05373-t002:** Material characteristics of inserted implants.

Implant	Density	The Thickness of Equivalent Dry Material (mm)	Tensile Strength (MPa)
Carbon fibres	200 g/m^2^	0.200	3450
Glass fibres	166 g/m^2^	0.160	2476
Aluminium	2.7 g/cm^3^	5	170

**Table 3 polymers-14-05373-t003:** Samples conditioning parameters.

Treatment Code	Climatic Conditions (Temperature, °C/Rel. Humidity of Air, %)	Time to Constant Weight (Days) (<0.1% Mass Change)	Equilibrium Moisture Content E.M.C. (%)
A	Standard 23 °C/50%	10	9 ± 0.5
B	Dry 23 °C/30%	5	6 ± 0.5
C	Humid 23 °C/70%	5	14 ± 0.5

## Data Availability

The data presented in this study are available upon request from the corresponding author.

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
