# Peer review of "Analysis of Bonding Mechanisms of Various Implants and Adhesives in Laminated Oak-Wood Elements"

_polymers, 2022, doi:10.3390/polym14245373_

Round 1

Reviewer 1 Report

L39 strike ‘the most’

L40 ‘are affordable and low in weight’

L43 strike ‘therefore’. The preceding sentences to not set up the following one; for example no mention of glulam until then.

L45  redundant mention of cross-section.

L75 strike ‘another’

L78-83j. This doesn’t seem needed to me.

L84. Planing – the use of a plane

L100. Just ‘oak’. Oak-wood is not common usage

L103 mechanism? Was this really investigated?

L107 rober

L126-129 this is background/justification and belongs in the previous section if needed – you already mentioned face milling.

L181 “Reference”

Results and discussion. In general, I think the “Results’ section is too long (about ½ the paper) and the discussion too short, in part because there is discussion in the Results section and in part because some of the results are more detailed than they need to be. I suggest 1) move the parts of the results that are in the discussion to the discussion, and 2) double-checking that all the results points need to be explicitly mentioned.

Author Response

Dear reviewer,

thank you for your comments. All the responds and further explanations are attached.

Author Response

Dear reviewer,

thank you for your comments. Attached please find all the responses and clarifications.
